# Booster Vaccination Decreases 28-Day All-Cause Mortality of the Elderly Hospitalized Due to SARS-CoV-2 Delta Variant

**DOI:** 10.3390/vaccines10070986

**Published:** 2022-06-21

**Authors:** Veronika Müller, Lorinc Polivka, Istvan Valyi-Nagy, Alexandra Nagy, Zoltan Szekanecz, Krisztina Bogos, Hajnalka Vago, Anita Kamondi, Ferenc Fekete, Janos Szlavik, Jeno Elek, György Surján, Orsolya Surján, Péter Nagy, Zsuzsa Schaff, Cecília Müller, Zoltan Kiss, Miklós Kásler

**Affiliations:** 1Department of Pulmonology, Semmelweis University, 1083 Budapest, Hungary; polivka.lorinc@outlook.com (L.P.); nagy.alexandra0718@gmail.com (A.N.); 2South-Pest Hospital Centre, National Institute for Infectiology and Hematology, 1097 Budapest, Hungary; drvnistvan@gmail.com (I.V.-N.); szlavikjanos@dpckorhaz.hu (J.S.); 3Department of Rheumatology, Faculty of Medicine, University of Debrecen, 4032 Debrecen, Hungary; szekanecz.zoltan@med.unideb.hu; 4National Korányi Institute of Pulmonology, 1122 Budapest, Hungary; bogos@koranyi.hu (K.B.); elekjeno@gmail.com (J.E.); 5Department of Cardiology, Department of Sports Medicine, Semmelweis University, 1122 Budapest, Hungary; vago.hajnalka@kardio.sote.hu; 6National Institute of Mental Health, Neurology and Neurosurgery, 1145 Budapest, Hungary; kamondianita@gmail.com; 7Heim Pál National Pediatric Institute, 1089 Budapest, Hungary; efekete@heimpalkorhaz.hu; 8Ministry of Human Resources, 1055 Budapest, Hungary; gyorgy.surjan@emmi.gov.hu (G.S.); miniszter@emmi.gov.hu (M.K.); 9Institute of Digital Health Sciences, Semmelweis University, 1094 Budapest, Hungary; 10Department of Deputy Chief Medical Officer II., National Public Health Center, 1097 Budapest, Hungary; surjan.orsolya@nnk.gov.hu; 11National Institute of Oncology, 1122 Budapest, Hungary; peter.nagy@oncol.hu; 12Department of Anatomy and Histology, University of Veterinary Medicine, 1078 Budapest, Hungary; 13Institute of Oncochemistry, University of Debrecen, 4012 Debrecen, Hungary; 14Department of Pathology, Semmelweis University, 1085 Budapest, Hungary; schaff.zsuzsa@med.semmelweis-univ.hu; 15National Public Health Center, 1097 Budapest, Hungary; muller.cecilia@nnk.gov.hu; 16Second Department of Medicine and Nephrology-Diabetes Center, University of Pécs Medical School, 7624 Pécs, Hungary; dr.zoltan.kiss.privat@gmail.com

**Keywords:** SARS-CoV-2 infection, delta variant, vaccine effectiveness, mortality, hospitalization, elderly, comorbidities

## Abstract

(1) Background: SARS-CoV-2 infections are associated with an increased risk of hospital admissions especially in the elderly (age ≥ 65 years) and people with multiple comorbid conditions. (2) Methods: We investigated the effect of additional booster vaccinations following the primary vaccination series of mRNA, inactivated whole virus, or vector vaccines on infections with the SARS-CoV-2 delta variant in the total Hungarian elderly population. The infection, hospital admission, and 28-day all-cause mortality of elderly population was assessed. (3) Results: A total of 1,984,176 people fulfilled the criteria of elderly including 299,216 unvaccinated individuals, while 1,037,069 had completed primary vaccination and 587,150 had obtained an additional booster. The primary vaccination series reduced the risk of infection by 48.88%, the risk of hospital admission by 71.55%, and mortality by 79.87%. The booster vaccination had an additional benefit, as the risk of infection, hospital admission, and all-cause mortality were even lower (82.95%; 92.71%; and 94.24%, respectively). Vaccinated patients needing hospitalization suffered significantly more comorbid conditions, indicating a more vulnerable population. (4) Conclusions: Our data confirmed that the primary vaccination series and especially the booster vaccination significantly reduced the risk of the SARS-CoV-2 delta-variant-associated hospital admission and 28-day all-cause mortality in the elderly despite significantly more severe comorbid conditions.

## 1. Introduction

Coronavirus disease (COVID)-19 severity is dependent on severe acute respiratory syndrome coronavirus 2 (SARS-CoV-2) variants of concern (VOC), the patient’s comorbidities, and their vaccination status [1,2,3,4,5,6,7]. The SARS-CoV-2 delta variant caused the fourth wave of the pandemic in Hungary starting from 13 September 2021 and lasting until 31 December 2021. Vaccination has been the utmost tool for reducing the risk of infection, hospitalization, and mortality [1,2,3]. The vaccination program has already been in place with a large proportion of the population having primary immunization (2× vaccinated by mRNA vaccines Pfizer-BioNTech or Moderna, or 2× vaccinated by Sputnik or Sinopharm, or 1× Janssen vaccinated), which has resulted in decreased infection and hospitalization rates in Hungary [1]. Differences in VOC and the falloff of immunity following baseline immunization led to the introduction of booster vaccinations in different countries including Hungary [8]. Infections from the delta VOC were of particular interest, as vaccination was already performed on a large proportion of the Hungarian population, and booster vaccinations were started as a result of observed immunity loss following primary immunization in this time period [3]. Elderly people (over 65 years old) represent the most vulnerable population, with significantly increased risk of hospitalization and mortality from SARS-CoV-2 VOCs. Age itself and multiple comorbidities are the highest risk factors contributing to an increased risk of COVID-19 and it is a more severe outcome [9,10,11,12,13]. There are insufficient data available about COVID-19 mortality caused by SARS-Cov-2 delta VOC in elderly patients needing hospital care according to their vaccination status [2]. Our main focus was to determine infection, hospitalization, and a 28-day all-cause mortality rate in the elderly with primary immunization and the additional protective effect of booster vaccination during the SARS-CoV-2 delta VOC pandemic in Hungary. Additionally, patient factors influencing hospitalization and 28-day all-cause mortality were assessed.

## 2. Materials and Methods

### 2.1. Study Population

This nationwide, retrospective, observational study examined SARS-CoV-2 hospitalization and 28-day all-cause mortality from the SARS-CoV-2 delta VOC, based on individual data. Infected cases between 23 August and 5 December 2021 were included, focusing on elderly patients hospitalized between 13 September and 30 November 2021, ensuring all cases to be included into the hospitalized cohort who had a SARS-CoV-2 positive result in the interval 21 days before or 5 days after hospital admission. In our study, “elderly” has been defined as being over 65 years old as it has often been used in medical research [14]. Data from the National Public Health Centre available between 22 January 2021 and 31 December 2021 were analyzed, allowing for all patients a minimum of a 28-day follow-up to assess all-cause mortality. Cases of SARS-CoV-2 delta VOC infection were reported on a daily basis using a centralized system via the National Public Health Centre. The report is based on (a) COVID-19-related symptoms identified by emergency team physicians or general practitioners and (b) a positive nucleic acid amplification test or rapid antigen test (RAT) reported by microbiological laboratories or the treating physician. RATs from the European Commission list were used [15]. The population was divided into groups based on vaccination status. These categories were: unvaccinated without prior infection (considered as reference), unvaccinated with prior infection (PCR or RAT positivity more than 120 days ago), vaccinated with only one dose, and primary and booster vaccinated. A person was considered primary vaccinated 14 days after receiving 2 doses of Pfizer-BioNTech, Moderna, Sputnik V, Astra Zeneca, Sinopharm, or a single dose of the Janssen vaccine. Booster vaccinations started from 1 August 2021 in Hungary and patients were included into the booster cohort 14 days following their booster vaccine. The booster included all available vaccines, with the suggestion to use mRNA vaccines following non-mRNA primary vaccinations. In Hungary, the mRNA booster was predominantly used (94.69% of all booster vaccines, assessed on 31 December 2021). The primary outcome was 28-day all-cause mortality following SARS-CoV-2 delta VOC-related hospitalization. The definition was based on World Health Organization recommendations and defined by the healthcare government in the National Social Information System [16]. Age, sex, and the four most frequent comorbidities—heart failure, type 2 diabetes mellitus (T2DM), chronic obstructive pulmonary disease (COPD), and malignancies—were analyzed according to vaccination status in the populations needing hospitalization and in the 28-day all-cause mortality subgroup. Analysis of the population by age groups was not performed due to the given life-expectancy in Hungary and due to low numbers in some of the subgroups by vaccination status [17].

### 2.2. Statistical Analysis

The analyzed populations (infected, hospitalized, died) were categorized by vaccination status at the date of infection and described by average age (±standard deviation), female/male ratio, and the prevalence of the 4 investigated comorbidities and the number of comorbid conditions. Data were analyzed using Student’s *t* and Chi-squared tests. *p* < 0.05 was considered statistically significant. To investigate the benefit of vaccination against the risk of infection, hospitalization, and 28-day all-cause mortality after hospitalization, relative risk reduction was calculated using the population that was unvaccinated and had no prior infection as a reference. The incidence rates were calculated for the analyzed populations from the timely mean population by vaccination status. From the incidence rates, relative risk reduction (RRR) and 95% confidence intervals (95% CI) were calculated utilizing the mid-p method from the epitools package in R [18]. The incidence rates were again calculated for the analyzed populations from the timely mean population by vaccination status. Using a mean population was to compensate for the change caused by the ongoing vaccination campaign. Due to this campaign, more than half of the primary vaccinated population received a booster vaccination during this period; hence, the populations defined by the vaccination status changed significantly during the study period. The mean populations were calculated as the average of the daily counts in the observation period. The number of patients at the start and end time point, as well as the calculated populations, are shown in Table 1.

As an additional calculation, we used a logistic multiple regression model for outcomes of hospitalization and all-cause mortality in 28 days after hospitalization in the infected population. The predicting variables were: age (as a continuous variable), sex (male/female), vaccination status (none vs. primary, none vs. booster), previous infection (yes/no), and comorbid conditions (present/not present). The result of the multivariate analysis is presented with odds ratios (OR) with corresponding 95% confidence interval (CI); for the multivariate analysis, SPSS version 27 was used [19].

## 3. Results

From the 1,984,176 elderly population (male: female = 769,477:1,214,699), people dying from any cause in the observation period, resulted in a small decrease of the total population included into the analysis. In the remaining population, 1,037,069 received the primary vaccination, while 587,150 received additional booster shots during the analyzed period and 299,216 remained unvaccinated. In the study period, 41,093 infections were registered, including 13,006 in unvaccinated people and 28,087 in vaccinated individuals. Hospitalization of elderly people associated with SARS-CoV-2 VOC was necessary in 0.56% of the total study population (n = 11,143), comprising 5087 unvaccinated and 6056 vaccinated cases. The risk of hospitalization in the primary vaccinated population was decreased by 71.55% and in the booster vaccinated population by 92.71% as compared to the reference unvaccinated patients. In hospitalized patients, 28-day all-cause mortality was analyzed for 3,996 cases. In primary vaccinated patients, the relative risk reduction of 28-day all-cause mortality was 79.87%, which was even better in the booster vaccinated group (94.24%), resulting in a decreased risk, respectively, as compared to unvaccinated individuals. While representing a very small proportion, unvaccinated but formerly infected individuals had a similar beneficial outcome as booster vaccinated patients. All studied populations and ratios are summarized in Table 2.

Hospitalized patient characteristics with the most prevalent comorbidities and including the number of comorbidities are summarized in Table 3.

While age did not differ, the primary and booster vaccinated populations included proportionally significantly fewer women. Hospitalized patients were significantly older and less frequently women and more often had malignancies in the booster vaccinated as compared to the primary vaccinated group. All four comorbid conditions were significantly more frequent in vaccinated as compared to unvaccinated populations, and multiple comorbidities were also significantly more prevalent in booster as compared to primary vaccinated patients. Table 4 summarizes patient characteristics of the 28-day all-cause mortality subgroup of previously hospitalized patients.

Patients were older in the 28-day all-cause mortality subgroups following hospitalization than the hospitalized total populations, and similarly, fewer women were affected in the primary and booster vaccinated groups. Heart failure, T2DM, and malignancies were more common than in the total hospitalized groups. All four comorbid conditions were significantly more frequent in vaccinated as compared to unvaccinated populations. Booster vaccinated patients were significantly older than primary vaccinated patients.

As an alternative balance for known confounders—like comorbid conditions and age—multivariate analysis confirmed that male sex, an increase in age, heart failure, COPD, T2DM, and malignancies have a significant negative effect for predicting hospitalization and 28-day all-cause mortality in the infected population (*p* < 0.000). On the other hand, primary and booster vaccinations sustained a significant protective effect (Appendix A).

The absolute distribution of vaccination variations of the primary and booster vaccinated populations at the beginning and at the end of the observation period are summarized in Appendix A.

## 4. Discussion

The SARS-CoV-2 pandemic led to the death of millions of people worldwide, and elderly people were the most frequently affected by this infection and its complications. In our study, we provided evidence that primary vaccination, and above all, booster vaccination, reduced the risk of infection, hospitalization, and especially 28-day all-cause mortality following hospital admission from SARS-CoV-2 delta VOC in this vulnerable population. In our cohort, infection risk was significantly lower in primary vaccinated (by 48.88%) and especially in booster vaccinated (by 82.95%) populations as compared to unvaccinated elderly. These data are in line with our previous results regarding data from the alpha VOC and with international studies, clinical trials, and real-word settings [1,3,12,20,21,22,23,24,25,26,27,28]. However, results from different studies should be carefully compared due to potential confounders including vaccine doses, vaccine types, and differences of VOCs [1,8]. Despite high overall primary immunization of the elderly population, an additional booster (mainly mRNA) did significantly decrease the hospital admission rate in this population. These data could be in line with the fact that elderly patients might become unprotected earlier as they have lower antibody levels after primary vaccination and an additional booster facilitates the production of antibodies [28,29]. Hospital admission risk was decreased by 92.71% in booster vaccinated as compared to unvaccinated patients, confirming data from previous research [2,3,12]. Although the actual number of vaccinated cases requiring hospitalization was slightly higher than those of primary and booster vaccinated cases during the study period, the proportion was significantly lower, because the total populations of primary and booster vaccinated and unvaccinated elderly individuals in Hungary did hugely differ (1,037,069; 587,150; and 299,216, respectively).

Heart failure, COPD, T2DM, and malignancy dominated as the most prevalent comorbidities in the vaccinated (especially in booster vaccinated) groups. According to our estimation in vaccinated cases requiring hospitalization, comorbid factors were represented significantly more frequently compared to our reference group, which means that patients in the primary and booster vaccinated groups tended to be more sick as compared to the reference. This serious burden may have contributed to a higher individual risk of infection, hospital admission, and mortality, which is also supported by previous data [10,11,30,31,32]. Additionally, these comorbid factors, male sex, and older age might partially limit vaccine effectiveness, which was also suggested by some previous studies [2,9,33,34]. The most important finding of our study is that the 28-day all-cause mortality of hospitalized patients with primary and booster vaccination was associated with significant benefit as compared to the unvaccinated population. Booster vaccination provided additional protective effects to primary vaccination. The 0.71% incidence rate of death in the reference population was significantly reduced by primary vaccination (0.14%), which was further improved by booster vaccination (0.04%). This reduction is even more impressive when considering that both vaccinated populations consisted of older patients with significantly more comorbid conditions, which was especially prominent in the booster group. Additionally, it is important to note that malignancy did affect nearly every fourth person in the primary and booster vaccinated groups, and malignancy-related mortality cannot be excluded when assessing all-cause mortality in these patients.

Past infection immunity against SARS-CoV-2 infection might provide protection against reinfections as shown by previous studies and also confirmed by our data [2,21,35,36]. However, due to the low number of patients with this outcome, this needs further investigations which is one of our future aims. Considering the sex-disaggregated data, women were more abundant in the elderly population, and they were similarly overrepresented in the unvaccinated population. In contrast, women needed hospital admission less frequently when vaccinated, and this ratio was significantly more pronounced in booster vaccinated patients. It is an interesting question whether this is due to differences in immune responses or whether it is simply the result that women tend to seek medical care earlier [33,34,37]. Age dependency is also important, as patients over 80 are at an even higher risk than patients between 65–79 years, and comorbid conditions have a high impact on the outcome [9,10,12,13,38,39,40].

Limitations of our data are mostly due to latent (non-reported) infections that might have altered the data of unvaccinated patients, as 16,176 previously unvaccinated patients (survivals of previous infection) showed similar protection to the booster population. The effect of prior infection was not investigated due to the low number of infected and hospitalized patients representing a much smaller population compared to the other categories. An additional limitation is that all patients were included into the hospitalized cohort with a positive SARS-CoV-2 result starting from 21 days before the admission, and this might result in a patient pool that were hospitalized mainly due to other reasons and not for coronavirus disease. All-cause mortality was used to exclude differences in the definition of COVID-19-related death. We would like to emphasize that we investigated all-cause mortality in a group which was hospitalized due to COVID-19, and this endpoint could be still interpreted as death at least partly related to the infection.

On the other hand, strengths of our study include its nationwide cohort, which represent data from an Eastern European country with a huge number of elderly people, focusing on the most vulnerable population group. It was unique in our methodology that multiple vaccinations were used and also comorbidities and their simultaneous appearance were examined in the light of hospitalization and 28-day all-cause mortality.

## 5. Conclusions

Primary and especially booster vaccinations contributed to the reduction of SARS-CoV-2 delta VOC-associated infections in the most vulnerable population of the elderly. Female gender and the lack of comorbidities proved to be a protective factor against severe infections. Effective vaccination campaigns as primary prevention has favorable public health effects and take burden off from healthcare systems. Booster vaccinations were the most effective in reducing 28-day all-cause mortality in hospitalized patients. Hence, timely vaccination campaigns might be encouraged to provide effective protection for the elderly against future SARS-CoV-2 VOC.

## Figures and Tables

**Table 1 vaccines-10-00986-t001:** The change in distribution of the population by vaccination during the study period (n).

	All	Unvaccinated without Prior Infection	Unvaccinated with Prior Infection	1 Dose	Primary Vaccinated	Booster Vaccinated
2021 September 13	1,984,176	312,315	18,126	40,962	1,377,768	234,977
2021 November 30	1,958,468	283,759	14,086	30,408	731,751	898,348
“Mean”	1,972,487	299,216	16,176	32,818	1,037,069	587,150

**Table 2 vaccines-10-00986-t002:** SARS-CoV-2 delta VOC infection, hospitalization, and mortality in the studied populations.

	Unvaccinated with No Prior Infection (Reference)	Unvaccinated with Prior Infection	1 Dose Vaccinated	Primary Vaccinated	Booster Vaccinated
Population (n)	299,216	16,176	32,818	1,037,069	587,150
Infected (n)	12,843	163	1034	22,757	4296
Incidence rate (%)	4.29	1.01	3.15	2.19	0.73
Relative risk reduction [95% CI] (%) As compared to reference		76.52 [72.6–80.01]	26.59 [21.79–31.17]	48.88 [47.75–49.97]	82.95 [82.35–83.54]
Hospitalized patients (n)	5064	23	339	4993	724
Incidence rate (%)	1.69	0.14	1.03	0.48	0.12
Relative risk reduction [95% CI] (%) As compared to reference		91.6 [87.38–94.68]	38.97 [31.86–45.48]	71.55 [70.41–72.65]	92.71 [92.12–93.27]
Hospitalized patients who died in 28 days (n)	2133	4	130	1488	241
Incidence rate (%)	0.71	0.02	0.40	0.14	0.04
Relative risk reduction [95% CI] (%) As compared to reference		96.53 [91.11–99.06]	44.43 [33.64–53.81]	79.87 [78.48–81.17]	94.24 [93.42–94.98]

CI: Confidence Interval.

**Table 3 vaccines-10-00986-t003:** Characteristics of SARS-CoV-2 delta VOC hospitalized patients.

	No Prior Infection (Reference)	1 Dose Vaccinated	Primary Vaccinated	Booster Vaccinated
	n = 5064	n = 339	n = 4993	n = 724
Average age (years)	76.67 ± 8.02	77.44 ± 7.76	76.67 ± 7.39	78.6 ± 7.62 ** ^#^
Female n (%)	3214 (63.47)	218 (64.31)	2609 (52.25) **	348 (48.07) ** ^#^
Comorbidities n (%)				
Heart failure	338 (6.67)	33 (9.73) *	491 (9.83) **	85 (11.74) **
COPD	405 (8)	28 (8.26)	537 (10.76) **	91 (12.57) **
T2DM	1372 (27.09)	116 (34.22) **	1866 (37.37) **	286 (39.5) **
Malignancy	533 (10.53)	53 (15.63) **	911 (18.25) **	170 (23.48) ** ^#^
Number of comorbidities n (%)				
None ^a^	2984 (58.93)	161 (47.49)	2153 (43.12)	271 (37.43)
1 ^b^	1598 (31.56)	135 (39.82)	2043 (40.92)	302 (41.71)
2 ^b^	402 (7.94)	34 (10.03)	643 (12.88)	124 (17.13)
3 ^b^	71 (1.4)	9 (2.65)	140 (2.8)	26 (3.59)
4 ^b^	8 (0.16)	0 (0) **	14 (0.28) **	1 (0.14) ** ^#^

COPD: Chronic obstructive pulmonary disease, T2DM: Type 2 diabetes mellitus. All data are given as n or average ± standard deviation if not otherwise stated. ^a^ None of the investigated comorbidities is present. ^b^ The number of investigated comorbidities is present. * *p* < 0.05 vs. unvaccinated; ** *p* < 0.01 vs. unvaccinated, ^#^
*p* < 0.05 vs. primary vaccinated.

**Table 4 vaccines-10-00986-t004:** Patient characteristics of the 28-day all-cause mortality subgroup following hospitalization.

	No Prior Infection (Reference)	1 Dose Vaccinated	Primary Vaccinated	Booster Vaccinated
	n = 2133	n = 130	n = 1488	n = 241
Average age (years)	78.43 ± 8.26	78.7 ± 8.15	78.6 ± 7.69	80.52 ± 8.06 ** ^#^
Female n (%)	1294 (60.67)	79 (60.77)	708 (47.58) **	108 (44.81) **
Comorbidities n (%)				
Heart failure	164 (7.69)	18 (13.85) *	180 (12.1) **	36 (14.94) **
COPD	173 (8.11)	13 (10)	160 (10.75) **	30 (12.45) *
T2DM	631 (29.58)	49 (37.69)	597 (40.12) **	99 (41.08) **
Malignancy	237 (11.11)	25 (19.23) **	285 (19.15) **	58 (24.07) **
No. of comorbidities n (%)				
None ^a^	1203 (56.4)	57 (43.85)	590 (39.65)	90 (37.34)
1 ^b^	700 (32.82)	48 (36.92)	633 (42.54)	92 (38.17)
2 ^b^	188 (8.81)	18 (13.85)	213 (14.31)	47 (19.5)
3 ^b^	39 (1.83)	7 (5.38)	45 (3.02)	11 (4.56)
4 ^b^	3 (0.14)	0 (0) **	7 (0.47) **	1 (0.41) **

COPD: Chronic obstructive pulmonary disease, T2DM: Type 2 diabetes mellitus. All data are given as n or average ± standard deviation if not otherwise stated. ^a^ None of the investigated comorbidities is present. ^b^ The number of investigated comorbidities is present. * *p* < 0.05 vs. unvaccinated; ** *p* < 0.01 vs. unvaccinated, ^#^
*p* < 0.05 vs. primary vaccinated.

## Data Availability

The original contributions presented in the study are included in the article; further inquiries can be directed to the corresponding author.

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
