# Peer review of "Booster Vaccination Decreases 28-Day All-Cause Mortality of the Elderly Hospitalized Due to SARS-CoV-2 Delta Variant"

_vaccines, 2022, doi:10.3390/vaccines10070986_

Round 1
Reviewer 1 Report
Population based studies on the impact of Covid Vaccination in the elderly are interesting. The authors compared the exposures 1)anamnestic infection 2) one dose 3) complete primary 4) booster dose with reference to no preceding vaccination/infection with respect to the outcomes infection, hospitalisation and all cause mortality within 28 after the infection.
While basically confirming the effectiveness of primary vaccination and booster, one findings was surprising: the excellent effect of a preceding infection.
Although the paper is written very clearly und the analysis is nicely shown, I have some questions and suggestions:
- Is the analysis based on individual or aggregated data?
- In tables 3 and 4 you basically show, that in hospitalised /dead persons the exposure (vaccination status) is associated with a number of risk factors.
- Why did you not adjust for that? T-Statistics are nice to keep things simple but simple models may allow for a better understanding of what is going on. With individual data this should be simple.
- The remarkable protective effect of a preceding infection needs more discussion. Should these persons be given a vaccine shot?
- Many different vaccines were used for primary immunisation – could you present a table with the proportions?
Author Response
We would like to express our thanks to the Reviewers for the careful evaluation of the manuscript and the helpful and constructive suggestions.
Reviewer 1:
Population based studies on the impact of Covid Vaccination in the elderly are interesting. The authors compared the exposures 1) anamnestic infection 2) one dose 3) complete primary 4) booster dose with reference to no preceding vaccination/infection with respect to the outcomes infection, hospitalisation and all cause mortality within 28 after the infection. While basically confirming the effectiveness of primary vaccination and booster, one findings was surprising: the excellent effect of a preceding infection. Although the paper is written very clearly und the analysis is nicely shown, I have some questions and suggestions:
- Is the analysis based on individual or aggregated data?
The analysis is based on individual data. We included this information into the Methods part of the revised manuscript (Line 83).
- In tables 3 and 4 you basically show, that in hospitalised /dead persons the exposure (vaccination status) is associated with a number of risk factors. Why did you not adjust for that? T-Statistics are nice to keep things simple but simple models may allow for a better understanding of what is going on. With individual data this should be simple.
As the original purpose of the study was to describe the whole elderly population of Hungary, we did not want to exclude any data from our calculations. This meant we could not use simple adjusting techniques such as case-control or propensity score matching. Weighting for the comorbid conditions would have been an option, however our access was limited to data concerning only some of the comorbidities (such as the ones described in the paper). To avoid false or misleading results we agreed to only work with raw data. Additionally, the new version of the manuscript contains a multivariate analysis as an alternative to balance for known confounders, like comorbid conditions and age. This was included as Supplementary Table 1. We have added a paragraph about it into the Statistical Analysis (Line 133-139): “As an additional calculation, we used a logistic multiple regression model for out-comes of hospitalization and all-cause mortality in 28 days after hospitalization in the infected population. The predicting variables were: age (as a continuous variable), sex (male/female), vaccination status (none vs., primary, none vs. booster), previous infection (yes/no) and comorbid conditions (present/not present). The result of the multivariate analysis is presented with odds ratios (OR) with corresponding 95% CI, for the multivariate analysis SPSS version 27 was used.”
These data were included as a part of the Results section (Line 188-195) : “As an alternative balance for known confounders- like comorbid conditions and age- multivariate analysis confirmed that male sex, increase in age, heart failure, COPD, T2DM and malignancies have a significant negative effect for predicting hospitalization and 28-day all-cause mortality in the infected population (p<0.000). On the other hand, primary and booster vaccinations sustained significant protective effect (supplementary Table S1).”
- The remarkable protective effect of a preceding infection needs more discussion. Should these persons be given a vaccine shot?
This question is relevant, so we are planning to additionally analyze these data. Our primary goal was to investigate the benefit of primary and booster vaccinations on the population with confirmed previous infection, however as we agree with reviewer that this is a relevant question we have added this important point in the Discussion section: Line 241-242: “…However due to the low number of patients with outcome this needs further investigations which is one of our future aims...”
- Many different vaccines were used for primary immunization – could you present a table with the proportions?
Thank you very much for the suggestion. We added a tables detailing the proportions of primary and booster vaccinations at the beginning and end of the observation period as Supplementary Table 2 and 3.

Reviewer 2 Report
I'd like to thank the Editors for asking me to review this very interesting paper which significantly contributes to enriching the literature on SARS CoV-2.
The result of the study is that primary vaccination series and especially booster vaccination significantly reduced the risk of SARS-CoV2 delta variant associated hospital admission and 28-day all-cause mortality in elderly despite significantly more severe comorbid conditions.
The authors should broaden the discussion by comparing their findings with these other articles:
Coppeta L, Somma G, Ferrari C, Mazza A, Rizza S, Trabucco Aurilio M, Perrone S, Magrini A, Pietroiusti A. Persistence of Anti-S Titre among Healthcare Workers Vaccinated with BNT162b2 mRNA COVID-19. Vaccines (Basel). 2021 Aug 25;9(9):947. doi: 10.3390/vaccines9090947. PMID: 34579184; PMCID: PMC8472926.
Coppeta L, Balbi O, Grattagliano Z, Mina GG, Pietroiusti A, Magrini A, Bolcato M, Trabucco Aurilio M. First Dose of the BNT162b2 mRNA COVID-19 Vaccine Reduces Symptom Duration and Viral Clearance in Healthcare Workers. Vaccines (Basel). 2021 Jun 17;9(6):659. doi: 10.3390/vaccines9060659. PMID: 34204252; PMCID: PMC8234325.
Zhao J., Yuan Q., Wang H., Liu W., Liao X., Su Y., Zhang Z. Antibody responses to SARS-CoV-2 in patients of novel coronavirus disease 2019. Clin. Infect. Dis. 2020;71:2027–2034. doi: 10.1093/cid/ciaa344. - DOI - PMC - PubMed
Author Response
We would like to express our thanks to the Reviewers for the careful evaluation of the manuscript and the helpful and constructive suggestions.
I'd like to thank the Editors for asking me to review this very interesting paper which significantly contributes to enriching the literature on SARS CoV-2. The result of the study is that primary vaccination series and especially booster vaccination significantly reduced the risk of SARS-CoV2 delta variant associated hospital admission and 28-day all-cause mortality in elderly despite significantly more severe comorbid conditions.
- The authors should broaden the discussion by comparing their findings with these other articles: (3 attached articles)
The Reviewer has a good point. After reading and discussing these 2 articles below we decided to include a paragraph for these references into the Discussion sections as follows:
Line 209-212: “These data could be in line with the fact that elderly patients might become unprotected earlier as they have lower antibody levels after primary vaccination and additional booster facilitates the production of antibodies”.
Coppeta L et al. Persistence of Anti-S Titre among Healthcare Workers Vaccinated with BNT162b2 mRNA COVID-19. Vaccines (Basel). 2021 Aug 25;9(9):947. doi: 10.3390/vaccines9090947. PMID: 34579184; PMCID: PMC8472926.
Coppeta L et. al. First Dose of the BNT162b2 mRNA COVID-19 Vaccine Reduces Symptom Duration and Viral Clearance in Healthcare Workers. Vaccines (Basel). 2021 Jun 17;9(6):659. doi: 10.3390/vaccines9060659. PMID: 34204252; PMCID: PMC8234325
Zhao J et al. Antibody responses to SARS-CoV-2 in patients of novel coronavirus disease 2019. Clin. Infect. Dis. 2020;71:2027–2034. doi: 10.1093/cid/ciaa344. - DOI - PMC - PubMed
In the opinion of the authors this article is out of the manuscript’s content as antibody responses were not included in our study. However, we are open to the suggestion of the Reviewer about putting this article in context.

Reviewer 3 Report
The primary objective of this article is to determine the rates of infection,
hospitalization, and 28-day all-cause mortality in primary vaccinated persons over 65 years of age and the additional protective effect of booster vaccination during the SARS-CoV2 delta VOC pandemic in Hungary.
The authors do not specify their calculation of the Relative Risk Reduction which seems to me to be just a calculation of the rate of increase calculated from the reference population.
The authors did not perform a multivariate analysis taking into account comorbidities.
The authors did not present the comorbidities of non-vaccinated patients who were previously infected. cf table 3.
In fact, the infection and mortality rates are very low in the different groups. Therefore, the conclusion that emphasizes only the rate of increase seems a bit artificial. This is especially true since in the end the previously infected people seem to be very protected. The authors do not really discuss this result in favor of natural immunity, which they discuss within the limits of their study. Another limitation is the fact that hospitalization and death cannot be related to covid 19.
I think that the calculation methods should be made explicit and that a multivariate analysis should be performed.
It seems to me that it would also be interesting to analyze the data by age groups. From an epidemiological point of view the definition of an elderly person is 75 years.
Author Response
We would like to express our thanks to the Reviewers for the careful evaluation of the manuscript and the helpful and constructive suggestions.
Reviewer 3:
The primary objective of this article is to determine the rates of infection, hospitalization, and 28-day all-cause mortality in primary vaccinated persons over 65 years of age and the additional protective effect of booster vaccination during the SARS-CoV2 delta VOC pandemic in Hungary.
- The authors do not specify their calculation of the Relative Risk Reduction which seems to me to be just a calculation of the rate of increase calculated from the reference population.
Thank you very much for pointing out the lack of description of the calculation of Relative Risk Reduction (RRR). We included a short addition in the new version of the paper in the statistical methods part, detailing that the RRR was calculated using R with the "epitools" package. The RRR is calculated with this formula exactly:
RRR = (CER-IER)/CER, Where CER is the “Event Rate in Control group” in which the control group is the reference population. IER is the “Event Rate in the Investigated group”
We have added a paragraph about it into the Statistical Analysis (Line 122-126).
- 2. The authors did not perform a multivariate analysis taking into account comorbidities.
The use of a multivariate model was an excellent suggestion to we performed addition analysis to investigate the effects of known confounders (age, main comorbid conditions) and decided to perform a calculation using multiple logistic regression for the outcomes (hospitalized/died) in the infected group. We used age (as a continuous variable), sex (male/female), vaccination status (none vs. primary, none vs. booster), and comorbid conditions (present/not present) as predicting variables. Data were included as Supplementary Table S1.
We have added a paragraph about it into the Statistical Analysis (Line 133-139):
“As an additional calculation, we used a logistic multiple regression model for out-comes of hospitalization and all-cause mortality in 28 days after hospitalization in the infected population. The predicting variables were: age (as a continuous variable), sex (male/female), vaccination status (none vs., primary, none vs. booster), previous infection (yes/no) and comorbid conditions (present/not present). The result of the multivariate analysis is presented with odds ratios (OR) with corresponding 95% CI , for the multivariate analysis SPSS version 27 was used.”
These data were also included as a part of the Results section (Line 188-195) : “As an alternative balance for known confounders- like comorbid conditions and age- multivariate analysis confirmed that male sex, increase in age, heart failure, COPD, T2DM and malignancies have a significant negative effect for predicting hospitalization and 28-day all-cause mortality in the infected population (p<0.000). On the other hand, primary and booster vaccinations sustained significant protective effect (supplementary Table S1).”
The results of this new calculation are presented in the Supplementary Materials.
- 3. The authors did not present the comorbidities of non-vaccinated patients who were previously infected. cf table 3.
We excluded previously infected unvaccinated elderly as only 23 patients were hospitalized and only 4 died from this group. Descriptive analysis of this disproportionately small group could be misleading and could divert attention from the main focus of our paper.
- 4. In fact, the infection and mortality rates are very low in the different groups. Therefore, the conclusion that emphasizes only the rate of increase seems a bit artificial. This is especially true since in the end the previously infected people seem to be very protected.
Using relative risk of reduction (RRR)* is an international statistical standard for assessing vaccine efficiency so with its use we aimed to increase the comparability of our data in the future. As an example RRR is used in the article below to evaluate different vaccines based on results of vaccine effectiveness studies: Olliaro P, Torreele E, Vaillant M. COVID-19 vaccine efficacy and effectiveness-the elephant (not) in the room. Lancet Microbe. 2021 Jul;2(7):e279-e280. doi: 10.1016/S2666-5247(21)00069-0. Epub 2021 Apr 20. Erratum in: Lancet Microbe. 2021 Jul;2(7):e288. PMID: 33899038; PMCID: PMC8057721. Additionally, we would like to point out that with the inclusion of absolute frequencies in the same table as incidence and RRR we intended to increase transparency.
*RRR = (CER-IER)/CER, Where CER is the “Event Rate in Control group” in which the control group is the reference population. IER is the “Event Rate in the Investigated group”
The authors do not really discuss this result in favor of natural immunity, which they discuss within the limits of their study.
As we answered for comment #3 of the 1st Reviewer we additionally discussed our intent of further investigating the reinfected population and the benefit of vaccines in this group. We have added this topic into the Discussion section: Line 241-242: “…However due to the low number of patients with outcome this needs further investigations which is one of our future aims…”
Another limitation is the fact that hospitalization and death cannot be related to covid -19.
We fully agree with the Reviewer. Mortality is a hard endpoint making analysis more valuable even with known limitations. This limitation of the Discussion section was completed as follows in Line 259-262: “All-cause mortality was used to exclude differences in the definition of COVID-19 related death. We would like to emphasize that we investigated all-cause mortality in a group which was hospitalized due to COVID-19, and this endpoint could be still interpreted as death at least partly related to the infection.”
- I think that the calculation methods should be made explicit and that a multivariate analysis should be performed.
As written in the answers for comment #1 and #2 we included these requested additions to the new version of the paper.
- It seems to me that it would also be interesting to analyze the data by age groups. From an epidemiological point of view, the definition of an elderly person is 75 years.
The definition of elderly was discussed heavily at the beginning of planning this study. We settled for the definition of elderly by the WHO for multiple reasons. One being this meant a greater study population, meaning a greater power to any conclusions. Two other notable reasons are the retiring age, which is currently 65 years in Hungary and the life-expectancy, which was 76.02 years in 2019. Both of these factors are weighting our perception that an “elderly” population in Hungary can be considered as people with age of 65 years and older. These additional data concerning the definition of elderly was included to the Methods part (Line 111-113): “Analysis of the population by age groups was not performed due to the lower life-expectancy in Hungary and due to low numbers in some of the subgroups by vaccination status.”
Round 2
Reviewer 3 Report
The authors have taken into account the requests for clarification
This manuscript is a resubmission of an earlier submission. The following is a list of the peer review reports and author responses from that submission.